# Barriers to the Prevention and Control of Hepatitis B and Hepatitis C in the Community of Southwestern China: A Qualitative Research

**DOI:** 10.3390/ijerph16020231

**Published:** 2019-01-15

**Authors:** Tingting Li, Shu Su, Yong Zhao, Runze Deng, Mingyue Fan, Ruoxi Wang, Manoj Sharma, Huan Zeng

**Affiliations:** 1School of Public Health and Management, Chongqing Medical University, Chongqing 400016, China; Lee712ting@126.com (T.L.); zhaoyong@cqmu.edu.cn (Y.Z.); dengrunze059@163.com (R.D.); fanmingyue129@163.com (M.F.); 15562577422@163.com (R.W.); 2Research Center for Medicine and Social Development, Chongqing Medical University, Chongqing 400016, China; 3The Innovation Center for Social Risk Governance in Health, Chongqing Medical University, Chongqing 400016, China; 4School of Public Health and Preventive Medicine, Faculty of Medicine, Nursing and Health Sciences, Monash University, Melbourne, VIC 3004, Australia; Ssu27@student.monash.edu; 5Department of Behavioral and Environmental Health, Jackson State University, Jackson, MS 39213, USA; Health for All, Omaha, NE 68124, USA; Health Sciences, Walden University, Minneapolis, MN 55401, USA; manoj.sharma@jsums.edu

**Keywords:** hepatitis B, hepatitis C, qualitative research, China

## Abstract

Objective viral hepatitis is a big challenge in China. However, few studies have focused on mapping the difficulties from a broader view. This study aimed to identify the barriers to the prevention and control of hepatitis B and hepatitis C in communities from the perspectives of hepatitis patients, residents, and healthcare providers. A total of 26 participants were recruited through purposive sampling. Data were collected by in-depth face-to-face interviews from September 2015 to May 2016 in two communities from Chongqing and Chengdu, China. A thematic framework was applied to analyze the qualitative data from the interviews. The critical factors of barriers to hepatitis prevention and control in the districts included poor cognition of residents regarding hepatitis B and hepatitis C, severe stigma in society, inadequate health education, and the provision of unsatisfactory medical services. Strengthening health education and improving services for treating patients with hepatitis are suggested to make further progress. A substantial gap remains between the need and currently available services for hepatitis patients and residents. Delivering quality prevention and control health services, improving health education, and reducing stigma in society are recommended to improve the prevention and control program for hepatitis B and C in communities.

## 1. Introduction

Hepatitis B and hepatitis C remain serious public health issues that affect 325 million people globally [1] and has led to approximately 1.34 million deaths annually [2,3,4]. Most of these deaths are caused by untreated chronic hepatitis infections, resulting in cirrhosis and liver cancer [2]. Chronic hepatitis B and C infections account for approximately two-thirds of all cases of liver cancer globally [2]. According to a 2017 hepatitis report [3], 1.75 million adults were newly infected with hepatitis C virus (HCV) in 2015, largely due to drug use via injections and unsafe injections in health care settings in certain countries [4], and the mortality caused by viral hepatitis continually increases [3]. In China, liver diseases affect approximately 300 million people, and these diseases are mainly caused by the hepatitis B virus (HBV) and HCV infections [5]. Each year, approximately 300,000 Chinese die from HBV-related liver cirrhosis and hepatic cellular cancer, accounting for 37%–50% of mortality worldwide [6]. The prevalence of hepatitis B and C is higher in China [7,8,9] when compared with other Asian countries such as Japan, Singapore, and Pakistan [10,11,12,13,14,15]. Recent studies have shown that the incidence of hepatitis B has decreased [16,17], but approximately 120 million people in China remain as HBV carriers [18]. Other studies have indicated a significantly descending trend of reported HBV incidence in younger generations, but a significantly increasing trend in the older population [19,20]. The prevalence of HCV infection has increased in recent years [19,21]. To date, the low coverage of testing and treatment of hepatitis B and C is a substantial gap that must be addressed globally [1,22,23,24,25]. At least 60% of liver cancer cases result from the late testing and treatment of HBV and HCV [1]. In 2015, 9% and 20% of HBV-positive and HCV-positive patients worldwide were aware of their diagnosis, and out of these diagnosed patients, the treatment coverage reached 8% and 7.4%, respectively [23,25]. Based on the National Disease Supervision Information Management System of China, the mean reported incidence of hepatitis B was 84.3 per 100,000 in China between 2005 and 2010, the viremic HCV prevalence of China was 0.7% (95% uncertainty interval 0·5–0·8) in 2015 [26]. In China, antiviral medications had been received by 58.2% and 66.6% of HBV and HCV-infected patients in 2011, respectively [27], and nucleotide analogs were the major antiviral medications prescribed for HBV-infected patients (most commonly adefovir dipivoxil and lamivudine). Ribavirin + pegylated interferon was prescribed for two-thirds of HCV-infected patients [27]. The achievement of the global goal of eliminating hepatitis by the year 2030 remains hindered by obstacles such as reducing new infections and mortality by 90% and by 65%, respectively, moreover, 90% of the people with HBV and HCV infections should be tested, and 80% of eligible patients should receive treatment [3].

Chongqing and Chengdu are located in Southwest China, which is an economically and less-developed area. Numerous researchers have investigated the epidemiology of HBV and HCV infection in South/Southwest China. Zhu et al. [21] reviewed the spatiotemporal epidemiology of viral hepatitis in China and reported the increasing incidence rate of hepatitis C in Southwest China. One study indicated that the mean reported incidence of hepatitis B in Southwestern China reached 75~125 cases/100,000 people between 2005 and 2010 [7]. Another study emphasized the relatively high prevalence of HBV infection in Western China [28]. Liu et al. [19] conducted an observational population-based study to investigate the epidemiology of hepatitis B and C in China from 2004 to 2014, and observed a significantly increasing trend of reported HBV incidences in eight provinces in Southern and Southwestern China. For the reported HCV incidences, the most rapid relative increase was observed in five provinces in this region: Chongqing, Sichuan, Guizhou, Yunnan, and Tibet.

Nowadays, access to HBV and HCV testing and treatment serves as an important determinant for eliminating hepatitis by 2030. Community health service centers should offer basic services to meet the needs of residents. However, to our knowledge, no study has explored the problems and barriers to the prevention and control of hepatitis B and C in communities from the perspective of a health system. This study fills the academic gap to a certain extent and provides new insights into delivering effective prevention and control of hepatitis B and C in communities. In-depth interviews were conducted to collect information on the actual demands, obstacles, and suggestions from the perspectives of healthcare providers, community residents, and patients with hepatitis B and C. 

## 2. Materials and Methods

### 2.1. Study Design

A qualitative study was conducted in the form of in-depth interviews with hepatitis patients, community residents, and healthcare providers from Chongqing and Chengdu, Southwestern China from September 2015 to May 2016.

### 2.2. Settings and Recruitment

To reach out to potential participants to be interviewed, we contacted one hospital, two communities, two community health service centers, three Centers for Disease Control (CDC) offices, and two community-based non-governmental organizations (NGO) that address hepatitis prevention and control-related issues in Chongqing and Chengdu. In total, we invited 26 people, and all agreed to attend the interview. Of the ten healthcare providers, six CDC workers were in charge of hepatitis prevention and control programs and possessed substantive experience in conducting related programs. All recruited participants met the participation criteria and provided informed consent to be interviewed. No remuneration was provided for their involvement in the study.

### 2.3. Data Collection

The semi-structured interview protocol was shaped by our research purpose and extensive literature review. Key questions were formulated according to the demographic information (name, age, occupation, disease, and treatment), and current situation, barriers, and suggestions for hepatitis prevention and control in the community. Before the main study, the questions were piloted, and modifications were made accordingly. One doctor from the community health service center, one community resident, and one patient with hepatitis B were interviewed before the main interview. All the piloted interviewees were not included in the formal investigation.

Two trained and experienced interviewers conducted the interviews. One male researcher (RD) headed the interviews, and the other female researcher (TL) recorded the key points. Each interview was conducted in a private and quiet room with only the interviewee and interviewer to guarantee confidentiality. Detailed preliminary communication was conducted to ensure that the interviewer and interviewee learned every detail (objective, content, research method, time, location, profit, and risk) of the interview. The privacy of the participants was maintained and the data obtained were used for research purposes only. All interviews were carried out in the local language/dialect and lasted for 20–40 min. Data collection and analysis should continue to a point when additional input from new participants no longer changes the researchers’ understanding of the concept, which is the point of data saturation [29].In this study, we recruited interviewees strictly in line with the information saturation principle, and we stopped the interviews when there was no more information about the research theme.

### 2.4. Data Analysis

The recorded interviews were first transcribed from verbatim into Mandarin Chinese. Second, two investigators read the transcripts repeatedly and highlighted recurring themes. With regard to the themes emphasized in the transcripts and topic guide used in the interviews, a thematic framework was established after a discussion among the research team members. Later, all transcriptions were imported into MAXqda V12.0 (VERBI Software GmbH, Berlin, Germany) and coded line-by-line according to the thematic framework. The analysis panel consisted of two researchers with qualitative research experience and who could use the English language with proficiency to ensure quality translation of the texts. Finally, the illustrative quotes used for the results were initially translated into English and then independently back-translated into Chinese by two researchers to assure the quality of translation and consistency. All files were analyzed using the grounded theory.

### 2.5. Ethical Approval

The study protocol was approved by the Institutional Review Board of Chongqing Medical University, China.

## 3. Results

### 3.1. Demographic Characteristics of Interviewees

The 26 participants included five hepatitis patients, six residents, ten healthcare providers, two community leaders, and three NGO workers. Most of the patients were infected with the CHB virus. The three NGO employees worked in a community health promotion organization and AIDS prevention and control organization (Table 1).

### 3.2. Factors Related to the Difficulties of Effective Prevention and Control of Hepatitis B and C in the Community

(1) Poor awareness and knowledge of HBV and HCV care among participants and strong stigma in the society

The participants generally showed a lack of knowledge about hepatitis B and C. Some hepatitis patients provided incorrect perceptions about HBV and HCV, with several views considering hepatitis B as a kind of inherited disease. Almost all of the interviewed residents showed indifference to hepatitis B and C and lacked the initiative to undergo screening for these infections (see Quotes 1 and 2 in Table A1).

Several workers from the CDC and communities indicated that the majority of residents and medical workers from community health service centers lacked knowledge about hepatitis B and C (see Quotes 3 and 4 in Table A1).

Serious social discrimination and stigma still surround hepatitis B and C. Most hepatitis patients reported experiencing discrimination in life or work to varying degrees because of their condition (see Quote 5 in Table A1).

Workers from community health service centers expressed that they had treated HBV patients who were discriminated severely by their family members (see Quote 6 in Table A1).

(2) Inadequate health education about hepatitis B and C in the community

Our data showed that the health education on viral hepatitis in the community did not currently meet the need of the residents. The routine health education activities provided by the community health service centers generally concerned high blood pressure, diabetes, and heart diseases. However, health education activities for the prevention and control of viral hepatitis were rare (see Quote 7 in Table A1).

Thematic lectures about viral hepatitis were only conducted during World Hepatitis Day, and residents who attended could only receive the leaflets printed uniformly. There was no information materials targeted to local needs (see Quote 8 in Table A1).

Conducting effective health education was a challenge for the community to some extent. Since the current health education activities were not very attractive, few residents wanted to participate in such activities.

In addition, healthcare providers were unaware of their capacity to offer related reading materials. Others attended these activities not for learning, but for the gifts that were rewarded to the participants. Thus, providing health education in the community residents resulted in frustration mostly because of the lack of cooperation and motivation from them.

In summary, current health education activities are unrelated to the needs of residents and patients with hepatitis in the community.

(3) Low-level services currently provided to treat patients with hepatitis B and C

Community health service centers currently provide unmet services to test and treat hepatitis B and C. The community health service center can provide a diagnosis or screening of hepatitis B, but no such services about hepatitis C were available. Professional treatment in this regard could also not be provided. In addition, community health service centers did not offer screening for high-risk groups thus far.

Understaffing is the most crucial cause of failure of the community to deliver quality services to treat patients with hepatitis B and C. In-service doctors deal with a considerable amount of daily work, and most of them are general practitioners or traditional Chinese medicine doctors. The doctors who specialize in public health feature a limited capacity to engage in this endeavor, and almost all in-service doctors possess very limited clinical experience of treating patients with hepatitis B and C. As a result, community health service centers must refer suspected hepatitis B and C patients to a larger hospital for a higher level for diagnosis and treatment. However, a deputy director of community health service center pointed out that the bidirectional referral between the community health service center and the higher-level hospitals should be strengthened and improved (see Quotes 9 and 10 in Table A1).

Two community health service centers indicated the insufficiency of hepatitis drugs to meet the basic demands of patients and that there was also a lack of equipment for diagnosis and treatment (see Quote 11 in Table A1).

From the perspective of hepatitis patients, as they cannot receive effective treatment from community health service centers, the expensive treatment becomes a problem if they are treated in larger hospitals. Patients also spend considerable time in registration, transportation, and finances including therapy charges, traffic expenditure, and the food and living expenses of patients and family members caring for patients.

Community health service centers perform no follow-up, tracking, nor management of patients with hepatitis B and C. The following are possible reasons for the failure of doctors to manage hepatitis patients. First, the government mandates no such process. Second, the workload is considerably excessive for the community health service centers under the current manpower resources as a large number of patients with hepatitis need to be potentially managed. Finally, most patients are lost to follow up, or cannot be contacted due to the lack of telephone number or their unwillingness for follow up work.

In addition to unqualified health services and insufficient policy support, most of the interviewed doctors expressed the shortage of special funds for treating patients with hepatitis B and C in the community.

### 3.3. Participants’ Suggestions to Improve the Prevention and Control of Hepatitis B and C in the Community

(1) Strengthen health education

Almost all of the interviewees expressed the urgent need to enhance health education in the prevention and control of hepatitis B and C in the communities. Initially, most of the interviewed patients identified their urgent need for health education related to the prevention and control of hepatitis (see Quote 12 in Table A1).

Additionally, the interviewed inhabitants suggested that the forms of health education should be improved such as online activities (see Quote 13 in Table A1).

Furthermore, the community workers expressed that the target population of health education should be young people as most of them receive higher education and feature remarkable receptivity (see Quote 14 in Table A1).

Most of the community staff interviewed expressed their approval and support for the work of community health activists who exhibited a general sense of empathy and altruism. The community staff members aimed to achieve strengthened communication and cooperation with community health activists to take advantage of their positive role in expanding health education and enhancing community mobilization (see Quote 15 in Table A1).

(2) Improve services for treating patients with hepatitis B and C

Two community leaders recommended the advancement of services for treating patients with hepatitis B and C in community health service centers. First, recruiting some public health professionals and continuing the professional development of in-service medical workers are important to improve the professional level of the medical personnel.

In high-risk groups, augmenting the prevention of hepatitis B and C in high-risk groups including revaccination of adults with antibody deficiency is imperative (see Quote 16 in Table A1).

Finally, the routine work of preventing and controlling hepatitis B and C in community health service centers can motivate community health activists, clinical family doctors, and public health doctors and nurses to achieve work efficiency.

## 4. Discussion

This study identified substantial barriers to the prevention and control of hepatitis B and C in the community in Southwestern China. First, the residents showed poor awareness and knowledge about hepatitis B and C; moreover, social stigma and discrimination against patients with hepatitis B and C were also observed and the limited health education could not reach the population effectively. Second, the community health service centers faced difficulties in providing professional diagnosis and treatment to hepatitis patients due to the lack of capable professionals and equipment. Furthermore, the bidirectional referrals between the community health service center and the high level hospitals needs to be significantly consolidated.

Our findings revealed that the residents, patients with hepatitis, and medical staff from community health service centers had limited knowledge and awareness of hepatitis B and C. Similarly, another cross-sectional study that investigated the knowledge and practices concerning hepatitis B among healthcare and public health professionals in China also indicated the inadequate knowledge of health professionals about HBV [30]. However, France, as one of the countries with more effective hepatitis care delivery in Europe [31], showed adequate knowledge about hepatitis B in the general population [32]; Nigeria also presented considerable awareness on HBV infection and its transmission among the studied community residents [33]. There are many factors contributing to the differences between China and France. First, the prevention and control of viral hepatitis has been considered a public health priority in France since the early 1990s, and by 2012, France had implemented a national hepatitis strategy plan for several years. Part of the French viral hepatitis strategy includes an annual mass-media campaign targeted at the general public as well as specific campaigns to raise awareness among healthcare workers and groups at high risk. Subsidized treatment is also an important measure to improve the access of hepatitis patients to health care services. Thus, we recommend that the government should take actions to improve the level of health literacy of the general population, and increase the financial subsidies of treatment for hepatitis patients. Thus, enhancing health education about viral hepatitis in the community is imperative. Given the rapid development of information technology and the popularity of smartphones, almost all participants owned a smartphone with social applications (apps). We can utilize apps such as WeChat to convey the knowledge of prevention and control of hepatitis B and C.

This research showed the severe social stigma and discrimination against hepatitis B and C in China. Varaldo et al. [34] investigated the role of stigma and discrimination in Brazil from the point of view of the patients. They observed that after diagnosis, these diseases affected the patients’ life; 24.6%, 23.8%, and 10.1% of patients with hepatitis experienced physical avoidance by their family, being uninvited by friends for events, and being dismissed in their daily lives, respectively. One study in Australia also revealed that few pieces of research currently address the stigma and/or discrimination in relation to hepatitis B [35]. Therefore, our future efforts should focus on addressing the stigma and discrimination against hepatitis.

In this study, we observed that poor health service was a serious barrier to the prevention and control of hepatitis B and C in the community. Other studies in Australia [36,37] showed similar findings including the need to improve the clinical management of people with CHB, to encourage the health system to screen people at risk, and to improve access to clinical services and the knowledge and communication skills of primary health care and community health service providers. Broad-scale HBV and HCV screening are required for a large proportion of patients with HBV/HCV, who are likely to develop severe complications and transmit infection [38]. However, classical virological tests require blood sampling by venipuncture, capacity for cold storage, specific infrastructure, equipment, and personnel training, and are often unaffordable in low- to middle-income countries [39]. Therefore, it is necessary to examine alternatives to classical HBV and HCV virological tests. For example, alternative model using point-of-care (POC) tests is increasingly being considered for HBV and HCV screening, diagnosis, and monitoring. POC tests are small devices that provide qualitative and quantitative determination of viral antibodies and antigens, which has proven to be cost-effective for HBV and HCV screening in other countries [40,41]. Overall, new testing approaches that facilitate easy and inexpensive access and linkages to care need to be explored and recommended in China.

Our study has several limitations that must be acknowledged. First, owing to the limited time, manpower, and other resources, a small number of subjects were included. Second, the study was conducted among a limited set of communities, which were purposefully selected to reflect a range of performance levels and geographical regions. Thus, the findings may not be generalizable to other areas of China, but may offer some insights into the current challenges in the prevention and control of hepatitis B and hepatitis C in the community.

## 5. Conclusions

The study showed the considerable gap between the urgent demand for effective hepatitis prevention and control services and the current service provision to the residents and patients. A number of barriers and problems such as unqualified staff members, poor awareness of the general population, and social discrimination limit the prevention and control of hepatitis in communities. Therefore, comprehensive health services including screening, diagnosis, standardized treatment, and follow-up, should be provided. Expanding health education, reducing stigma in society, and providing supportive policies and funding from the government are also recommended.

## Figures and Tables

**Table 1 ijerph-16-00231-t001:** The socio-demographic characteristics of the interviewees.

Interviewee	Age	Sex	Location
Patient-1, Hepatitis C, under treatment	59	Male	Chengdu
Patient-2, Chronic hepatitis B, under treatment	49	Male	Chengdu
Patient-3, Chronic hepatitis B, under treatment	45	Female	Chengdu
Patient-4, Chronic hepatitis B, under treatment	46	Male	Chongqing
Patient-5, Chronic hepatitis B, under treatment	62	Female	Chongqing
Resident-1	61	Female	Chengdu
Resident-2	57	Female	Chongqing
Resident-3	62	Male	Chengdu
Resident-4	80	Female	Chongqing
Resident-5	43	Female	Chengdu
Resident-6	56	Female	Chengdu
CHC-1, Community Health Service Center Manager	33	Male	Chengdu
CHC-2, Community Health Service Center Manager	48	Female	Chongqing
CHC-3, Community Health Service Center worker	42	Female	Chongqing
CHC-4, Doctor from community health service center	27	Male	Chengdu
Community leader-1, Community manager	38	Female	Chengdu
Community leader-2, Community manager	54	Male	Chengdu
CDC-1, Infectious Disease Prevention and Control Section	49	Female	Chongqing
CDC-2, STD/AIDS Prevention and Control Department	39	Female	Chongqing
CDC-3, Department of Immunization Planning	37	Female	Chongqing
CDC-4, Infectious Disease Prevention and Control Department	42	Male	Chongqing
CDC-5, Infectious Disease Prevention and Control Department	55	Male	Chengdu
CDC-6, Infectious Disease Prevention and Control Department	38	Female	Chengdu
NGO-1, Community health promotion organization	54	Female	Chongqing
NGO-2, AIDS prevention and control organization	50	Male	Chongqing
NGO-3, AIDS prevention and control organization	37	Male	Chongqing

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
