# Peer review of "Barriers to the Prevention and Control of Hepatitis B and Hepatitis C in the Community of Southwestern China: A Qualitative Research"

_ijerph, 2019, doi:10.3390/ijerph16020231_

Round 1

Reviewer 1 Report

This study by Li T et al focusses on understanding the barriers to prevention and control of Hepatitis B and C in the communities of Chongqing and Chengdu in China through a qualitative survey of a subset of individuals within the community. The strongest criticism of the study - that even the authors acknowledge - is the limited number of subjects in this study. Apart from that, I encountered major concerns regarding the collection of data and how the data is presented in this article. Please read my specific comments below. 

1) Line 19-20: The authors claim that '26 participants were recruited through purposive samspling until data saturation was achieved'. There is no clear evidence provided in the methods of this article about how data saturation was achieved. 

2) Even though the study has 26 participants, Table 1 only lists data from the five hepatitis patients. Information about other interviewees is not listed. Also, basic demographic data is missing from the table such as - sex, marital status, number of children, birthplace, vaccination status, etc, which is basic information for any qualitative or quantitative survey study. 

3)  A lot of the survey results like patient comments can be tabulated to make the article easier to read. As of now, the results section just flows through as one giant section with patient comments mixed with author comments which made this article very hard to read for me.

Finally, the article has a lot of typos and grammatical errors throughout. I request the authors to thoroughly proofread the article to minimize such errors. 

Author Response

Dear reviewer,

Thanks for your kind consideration and insightful comments about my manuscript. We have studied your comments carefully and have made correction which we hope meet with your approval. According to the comments from reviewers, we have revised the result and discussion section mainly. Enclosed please find our revised manuscript where we addressed all the comments suggested by the reviewers. And we thank the journal for giving us this opportunity to revise this manuscript.

Best regards

                                                                             Yours sincerely,

                                                                              Tingting Li

Reviewer 2 Report

This article describes a qualitative research in China focused in HCV and HBV infection. The manuscript is well written and it merits publication.

Author Response

(The authors gave the same response as above.)

Reviewer 3 Report

This is an interesting manuscript exploring different stakeholders’ view on needs and opportunities for improvement of hepatitis care delivery in selected areas of Southern China. The authors provide valuable insights on the current situations and recommendations on future improvements. The manuscript would benefit from some revisions, see suggested inputs below.

Abstract – no background is reported in the abstract. I would suggest to add one sentence to state the relevance of the issue. If space is lacking the authors could consider rewording the conclusions as the two sentences are somewhat redundant.

Introduction

Lines 48-59 – The comparison between the global context and situation in China is interesting, however the introduction would benefit from some additional details on China: e.g. what is the prevalence of HCV in China? what is the testing coverage or proportion of diagnosed in China? When proportion of treated individuals is mentioned, there is no indication as to what is the denominator, year of observation or regimen used at least for HCV (i.e. DAAs).

Line 67 – please check the rate: 75~125 cases/100,1000?

Line 81 – Perhaps instead of “ordinary people” consider “health care users” or “community residents”

Methods

Line 92-94 – this would be better placed in the results section

Line 103 – perhaps specify who piloted the interviews

Line 116 – research “team” members

Results

General comment: The results section would improve if the findings were organised by stakeholders’ groups to emphasize differences and similarities in views and perceptions. The tone is generally too blunt and may need to be softened at points (see additional comments below) to better match the robustness and representativeness of the data available. The authors would need to use inverted comas to identify quotes.

Line 130 – consider change “afflicted” into “infected”

Table 1 – It would benefit from inclusion of details on all interviewees, possibly stratified by stakeholder group

Lines 141-2 and following instances – Use inverted comas to identify citations

Line 153 – “the best time for treatment has always been delayed”, unclear, please rephrase using e.g. early treatment

Line 155 – consider changing “incidence” into “occurrence” or simply delete the word

Line 169 – perhaps rephrase to be less categorical: e.g. our data show …. does not currently meet the need….

Line 173 – Would be better to change “publicity” into “attention” or “information” or similar

Line 177 – “produced and printed uniformly by China” does this mean there is no information materil targeted to local situation/needs? If so please state that instead of mentioning who the producer/distributor is

Lines 182-4 – please rephrase as the sentences are not clear. Also the messages are too categorical and may need to be softened and appropriately tuned to the robustness of the data available

Lines 192-7 – as above please consider rephrasing

Line 199 -  “Inaugural doctors” unclear

Lines 203-204– perhaps expand further on the current effectiveness of referral systems not only from the point of view of the patients (lines 215-216) but also of the service providers.

Discussion

General comment: the discussion touches on key concepts identified through the research study, but may benefit from some additional insights (see comments below). Also the authors could consider mentioning opportunities for improvement of care pathways, including decentralisation of treatments. There are few good examples on this from Australia

Lines 270-280 – As above some of the sentences are too blunt and should be toned down – the identification of a challenge could open the way for future changes, hence would be better presented in a more positive way.

Lines 285-288 – This paragraph presents an important concept, i.e. that a health care services impact on the population is unavoidably linked to the level of health literacy and disease awareness of that population. Yet it could be further expanded, perhaps also addressing in more details what may be the causes of such differences between China and other countries taken as examples. I would refrain from stating that France has the best hepatitis health care in Europe, perhaps “one of the more effective” is more appropriate.

Lines 301-314 – this paragraph explores the fundamental concept of expanding HBV/HCV testing and decentralising it to e.g. community health centers. Instead of listing the target groups proposed by the US guideines, the authors could consider digging more into the opportunities and feasibility of doing so also in China, e.g. presenting successful examples from elsewhere and referring to the opportunities offered by new POC testing devices.

Lines 318-19 – Consider rephrasing as “the findings may not be generalizable to other areas of China but may offer some insights on the current challenges in…..”

Author Response

(The authors gave the same response as above.)

Round 2

Reviewer 1 Report

The authors have made sufficient changes to the manuscript.